

# Evaluating simulations of ship tracks in a high-resolution model

Anna Tippett[1], Paul R. Field[2,3], and Edward Gryspeerdt[1]

[1]Department of Physics, Imperial College London, London, UK
[2]Met Office, Exeter, UK
[3]School of Earth and Environment, University of Leeds, Leeds, UK

**Correspondence:** Anna Tippett (a.tippett22@imperial.ac.uk)

**Abstract.** Clouds, and in particular their interactions with aerosols, remain a major source of uncertainty in climate projections, due to the wide range of scales over which cloud processes act on. This uncertainty limits our capability to simulate potential solar radiation management strategies, such as marine cloud brightening (MCB). A good natural analogue for investigating MCB is analysis of ship tracks, as these phenomena mimic the intended effect, and allow for us to investigate

time evolving aerosol perturbations. In this study, we use a real case of ship tracks as a natural experiment to model the time evolution of aerosol perturbations to marine stratocumulus clouds, evaluating model performance through comparisons with satellite observations. We evaluate our model simulations against three criteria, in order to ascertain whether this model is suitable for simulating MCB accurately. Our findings highlight a key deficiency in activation parameterisations when simulating high aerosol concentrations - such as those expected in MCB scenarios. While the model can replicate the mean cloud prop-

erties within ship tracks, it struggles to capture their temporal evolution. Specifically, in precipitating clouds, enhancements in droplet number concentration ($N_d$) and liquid water path (LWP) are overestimated and persist too long. This discrepancy between model and observations is linked to excessive model sensitivity to aerosol loading in precipitating conditions, leading to unrealistically easy suppression of drizzle, and ultimately resulting in simulated ship tracks which overestimate the cooling effect in these cases. We identify scenarios in which current formulations of parameterisations are not suitable for use in

simulating high-concentration aerosol perturbations, such as MCB, and scenarios in which models are more capable.

## 1 Introduction

A large portion of the uncertainty in estimates of the effective radiative forcing from aerosol-cloud interactions originates from poor model representation of clouds (Smith et al., 2020). Clouds depend on processes operating at micron scales, rendering their explicit representation unfeasible in most computationally viable resolutions of weather or climate models. Consequently,

cloud processes must be parametrised. These parameterisations define the process rates, which describe how much physical properties change between each model time step. Uncertainty in these parameterisations leads to considerable uncertainty in model estimates of the radiative forcing from aerosol-cloud interactions, with a large portion of this uncertainty stemming from the amount of liquid cloud (Zelinka et al., 2014). Addressing and reducing this uncertainty is crucial for enhancing the accuracy of future climate projections (Andreae et al., 2005). Better representation of clouds in regional weather forecasting

models would improve predictability of clouds and the onset of precipitation (Field et al., 2023). This uncertainty is important



for marine cloud brightening (MCB), a proposed solar radiation management (SRM) strategy aimed at reflecting more sunlight by intentionally seeding clouds with aerosols (Diamond et al., 2022; Wood, 2021; Salter et al., 2008; Zhang and Feingold, 2023). Uncertainty in model representation of high-concentration aerosol-cloud processes limits our capability to simulate MCB experiments, and makes the design of suitable field trials difficult.

To assess the longer term impacts of MCB on our climate, it must be incorporated into climate model simulations. One suggested approach involves the use of regional models to accurately simulate the localised effects of MCB, and use this to parametrise its impact in coarser resolution global climate models (Feingold et al., 2024). While large-scale turbulence is resolved in these models, many of the same parameterisations are used in GCMs (e.g., for activation or precipitation processes). Therefore, it is critical that we have certainty in our regional representations of MCB. Robust evaluation frameworks are needed

to assess the realism of regional models, not only in terms of short-wave (SW) radiative forcing but also with respect to the representation of key processes.

    Ship tracks, the narrow cloud features formed by aerosol emissions from ships, and offer a valuable natural experiment for studying the interactions that would occur in intentional MCB (Christensen et al., 2022; Diamond et al., 2022). Aerosols in the ship plume act as cloud condensation nuclei, causing the cloud to have more, smaller, droplets which reflect more incoming

solar radiation. This is known as the "Twomey" effect, and is well documented on short timescales (Twomey, 1974, 1977; Ferek et al., 1998; Hobbs et al., 2000; Ackerman et al., 2000; Feingold et al., 2003; Penner et al., 2004; Segrin et al., 2007; Christensen et al., 2022). The brightening observed in ship tracks mimics the intended effect of MCB, offering a real-world analogue for testing the feasibility of such interventions. Ship track studies provide information about aerosol-cloud interactions in this marine context, and can be useful they can be in assessing the behaviour of models that are intended for use in simulating

MCB.

    After the initial droplet number ($N_d$) perturbation, clouds respond to aerosol perturbations over longer timescales through effects known as 'cloud adjustments'. Changes in liquid water path (LWP) - the amount of liquid in a cloud column - are particularly important because they influence not only cloud radiative properties but also precipitation processes. A precipitation suppression effect (Albrecht, 1989; Rosenfeld, 2000), where smaller cloud droplets take longer to form precipitation

(increasing LWP), tends to result in additional cooling. Alternatively, smaller droplets can also enhance the mixing of dry air above cloud-top into the cloud (a process known as entrainment) and reduce LWP, resulting in a warming effect (Ackerman et al., 2004; Bretherton et al., 2007). This bidirectional response in the LWP can depend on the initial conditions of the unperturbed cloud (Han et al., 2002; Ackerman et al., 2004; Michibata et al., 2016; Toll et al., 2017, 2019; Gryspeerdt et al., 2019a; Possner et al., 2020; Glassmeier et al., 2021; Zhang et al., 2022; Fons et al., 2023), whilst also being controlled by covarying

meteorology (Goren et al., 2025).

    There is considerable uncertainty in both the sign and magnitude of adjustments (Glassmeier et al., 2021), which is relevant for MCB purposes, since adjustments involving decreasing the LWP could offset the intended cooling impact. Part of this uncertainty stems from both the difficulty in making observations of aerosol-cloud processes, and isolating the impact of aerosol from background meteorology (Christensen et al., 2022). Cloud adjustments are inherently time-dependent processes

(Gryspeerdt et al., 2021), yet many observational studies do not consider the time evolution of an cloud response, making



it difficult to observe the processes occurring. This limits our modelling capability of these interactions, since without time-resolved observations, it is difficult to capture the time dependence for instantaneous/short timescale injections (such as aerosols or MCB).

Simulating ship tracks provides an opportunity to simulate the intended effects of MCB, and evaluate a model's ability to represent key processes realistically. By simulating ship tracks we can disentangle aerosol effects from meteorological variability because the aerosol source is known and relatively localised. This is possible since we can use the region neighbouring the ship track as an unperturbed 'control' region (Segrin et al., 2007), which is a best estimate for what the cloud would have looked like if no ship was there. This makes them ideal for isolating the causal impact of aerosols on cloud properties and separate meteorological co-variations (Goren et al., 2025). Additionally, we can view ship tracks as linear formations of independently perturbed clouds (Kabatas et al., 2013), thereby allowing us to infer information about the time evolution of the aerosol perturbation (Gryspeerdt et al., 2021). This helps us evaluate model process representation, such as the activation of cloud droplets (which occurs on short timescales), or the autoconversion of cloud droplets into rain droplets (occurring on longer timescales). In order to answer logistical questions relating to MCB, accurate simulations of ship tracks are vital, however we must be certain that model simulations produce the correct answer for the right reasons. This calls for in depth evaluation of the representation of model processes, not just the final forcings. Ship tracks provide an avenue to do this.

Previous efforts have been made to simulate ship tracks in atmospheric models, in order to assess model performance (Wang and Feingold, 2009; Possner et al., 2015, 2018; Berner et al., 2015). Berner et al. (2015) and Wang and Feingold (2009) utilise fine resolution LES models to investigate the effect of aerosol perturbations on marine stratocumulus, but cannot make direct comparisons to observations since they are not real cases of ship tracks. Possner et al. (2015) use observations of a real case of ship tracks, yet the simulated ship locations are prescribed and not from the actual ships that caused the observed ship tracks.

In this work, we produce simulations of a real case of ship tracks, using ship emissions locations from ship Automatic Identification System (AIS) data. We utilise a double-moment cloud microphysics scheme - Cloud AeroSol Interacting Microphysics (CASIM; Field et al., 2023) in the Met Office Unified Model (UM) in regional configuration, and compare directly to satellite observations. We infer changes in cloud properties and the processes involved from changes in $N_d$ and LWP, and the time evolution of these responses. We evaluate our simulations according to the three criteria described in Section 2.1, in order to ensure the proper change in cloud radiative effect is captured via the correct microphysical processes. In order to properly simulate MCB impacts we must simulate the correct behaviour, and for the correct reasons.

Through evaluating our simulated ship tracks against these criteria, we reveal issues in current model parameterisations when modelling high-concentration aerosol perturbations, such as those that are expected in MCB. We identify scenarios in which current formulations are more suitable for simulating MCB, and scenarios in which further work is needed before models can be considered credible. Furthermore, we make suggestions on experimental design, such that analysis and quantification of the aerosol effect is least uncertain.





## 2 Methods

### 2.1 Evaluating ship tracks

1. ***Accurate representation of changes in CRE.*** The cloud radiative effect (CRE) in overcast scenes is largely dependent on the inside ship track $N_d$ and LWP values (Possner et al., 2015), therefore we want to ensure our model is producing simulations with the correct numerical values inside the ship track. Due to difficulties in obtaining perfect background simulations, our first criteria considers solely the absolute values where the aerosol perturbation has been applied, since this is ultimately where it is key to get the behaviour correct.

2. ***Accurate time evolution of the CRE.*** This criteria enables us to evaluate the lifetime of the aerosol effect and therefore the associated radiative forcing. Again, this effect is driven by changes in LWP and $N_d$. The model sensitivity to aerosol can be evaluated, as the enhancement from the unperturbed (control) state is taken into account. Ship tracks provide a method for obtaining time evolution even in snapshot satellite imagery (Gryspeerdt et al., 2021), therefore information about the time-evolution of the aerosol effect can be made.

3. ***Sensitivity in the $N_d$ response to initial conditions.*** Previous studies have shown that the cloud response is sensitive to the initial conditions of the cloud (Ackerman et al., 2004; Michibata et al., 2016; Toll et al., 2017, 2019; Gryspeerdt et al., 2019a; Possner et al., 2020; Glassmeier et al., 2021; Zhang et al., 2022; Fons et al., 2023). In order for our model representation of clouds to be accurate this sensitivity must be reflected in the model. We expect to see different responses in precipitating and non-precipitating conditions, and this is investigated in our model simulations. In order for the model
to reproduce the correct response for the right reasons, this dependence on the initial cloud conditions should be satisfied. This should confirm that the model is sensitive to the correct model processes.

### 2.2 Model configuration

Our case study is simulated using the Met Office Unified Model (UM, version 13.0; Brown et al., 2012), in the Regional Atmosphere and Land (RAL) configuration 3.1 (Bush et al., 2020, 2023, 2025). The 2625 km by 1125 km domain is centred
on (40ºN, 135.25ºW) with a horizontal resolution of 1.5 km. There are 90 levels up to 40 km, with 16 levels below 1 km. The model time step is 75 s, and the lateral boundary conditions to the nested regional model are provided hourly by the UM global model at N216 resolution ($\approx$ 60 km) in Global Atmosphere (GA) 6.1 science configuration (Walters et al., 2017). Our simulations are initialised from global model analysis at 0000Z on the 11th July 2018 and run for 48 hours.

To properly simulate the aerosol-cloud interactions in a cloud resolving model, a double moment cloud microphysics scheme
is recommended (Morrison et al., 2009; Igel et al., 2015). Double-moment microphysics schemes prognose both the mass and number concentration of each hydrometeor species (e.g. Ferrier, 1994; Seifert and Beheng, 2006; Morrison and Gettelman, 2008; Taufour et al., 2018). A double-moment microphysics scheme allows $N_d$ to be recalculated at each time step, and therefore can subsequently modify process rates (such as autoconversion), impacting the cloud evolution over time (Field et al., 2023). In single moment schemes, the number concentration of each species (from which the process rates are derived)



are either constant or diagnosed from other meteorological parameters using empirical relationships. These schemes, whilst computationally cheaper, can fail to accurately represent the indirect effect from aerosol-cloud interactions (Gordon et al., 2020).

For cloud microphysics, we employ the Cloud AeroSol Interacting Microphysics (CASIM) scheme (Shipway and Hill, 2012; Field et al., 2023). CASIM is a double moment scheme with five hydrometeor species (cloud liquid, rain, ice, snow, and
graupel) in which both mass and number concentration mixing ratios are prognosed.

Our configuration of CASIM allows $N_d$ to be derived from aerosol (rather than prescribed), and we couple it to the United Kingdom Chemistry and Aerosols (UKCA) aerosol microphysics scheme. UKCA is a full prognostic two moment aerosol microphysics scheme, based on GLOMAP (Mann et al., 2010), and provides both aerosol mass and number information for the activation of cloud droplets. To simplify our simulations, we only consider aerosol only in the accumulation mode, and
initialise our simulation with a constant sulfate aerosol number concentration of 200 cm$^{-3}$, characteristic mass $3 \times 10^{-18}$ kg, and aerosol diameter of $\approx 0.1 \mu$m, which is typical of sulphate aerosols (Noone et al., 2000). The aerosol activation scheme used is that of Abdul-Razzak and Ghan (ARG; 2000).

### 2.3   Control run

Using the model configuration detailed above, our simulation is run for 48 hours from 0000Z 11th July to 0000Z 13th July. In
our *control* run, no ship emissions are added to the model, and we use this simulation to characterise our "unpolluted" cloud. Typically, in ship track observations, we have no pure "control" cloud, and therefore we must treat a region outside of a ship track as being representative of what the cloud would have looked like if there was no aerosol perturbation present. In the model case, however, we can simply turn ships on and off to isolate the aerosol impact on the cloud.

The aerosol number concentration ($N_a$), cloud droplet number concentration ($N_d$), liquid water path (LWP), surface rain rate
(RR), and top of atmosphere outgoing short wave radiation (SW) for the *control* run 44 hours after initialisation can be found in the left-hand column of Fig. 1. Fig. 1a demonstrates how the coupled UKCA aerosol scheme allows depletion of aerosol in locations where there is precipitation, and the land sources of aerosol are the most significant over the course of the simulation. The coupled UM-CASIM configuration is able to reproduce open/closed cellular convection in this marine stratocumulus, with a drizzling scene (surface rain rates on average of roughly 0.5 mm/hr). Overpasses from the CloudSat Cloud Profiling Radar
(Stephens et al., 2008; Tanelli et al., 2008) verify this, with observations of pockets of precipitating clouds (with maximum surface rain rate of roughly 2 mm/hr) throughout the simulation run time (see Fig. S1).

With our initialised background aerosol field of 200 cm$^{-3}$, we produce a relatively clean cloud droplet background of roughly 50 cm$^{-3}$. This is largely in line with that seen in satellite observations (Fig. 5a,c), however we note that two large sources of cloud droplets are missing from our model simulation. This is likely due to our initialisation from a constant aerosol
field, and therefore we are missing these sources of aerosol in our simulation.





**Figure 1.** Left column: control simulations of 12th July 2018 at 20:00 without ship emissions turned on. Panel **(b)** contains the locations of the ships simulated during this study, up until the time step of this figure. Right column: the same simulation with shipping emissions on. Variables shown are aerosol number concentration ($N_a$), cloud droplet number concentration ($N_d$), liquid water path (LWP), surface rain rate (RR), and top of atmosphere outgoing short wave radiation (SW).





| Ship Label | Type | $lat_i$ (deg) | $lon_i$ (deg) | $lat_f$ (deg) | $lon_f$ (deg) | $v$ (knots) | $\frac{dN}{dt}$ (s$^{-1}$) |
|------------|------|------|------|------|------|------|------|
| A | Container | 38.768 | 234.396 | 44.762 | 219.362 | 19.895 | $5 \times 10^{14}$ |
| B | Container | 37.798 | 237.677 | 41.639 | 222.035 | 20.231 | $5 \times 10^{14}$ |
| C | Container | 37.796 | 237.686 | 44.738 | 222.516 | 20.179 | $5 \times 10^{14}$ |
| D | Container | 43.156 | 212.417 | 38.228 | 224.687 | 16.427 | $5 \times 10^{14}$ |
| E | Container | 43.337 | 210.858 | 36.580 | 227.716 | 22.534 | $5 \times 10^{14}$ |

**Table 1.** Start and end locations of 5 container ships, with associated velocities, emission rates. Ships are initialised at 00:00Z 11th July 2018, and allowed to travel for the 48 hours of the simulation.

## 2.4   Simulating ship tracks

In order to simulate the shipping impacts on clouds, we add five container ships to our simulation in our *ships* run. We use ship locations identified in AIS data and matched to observed shiptracks from 11th-12th July 2018, to add realistic moving sources of aerosol (Smith et al., 2015). Details of ship start/end positions, velocities, and ship type can be found in Tab. 1. These ships

were selected because they produced some of the most visible ship tracks within this domain, and covered both directions of travel across the domain.

Aerosol is added in a model level at 10 m above sea level and immediately dispersed throughout the 1.5 km by 1.5 km grid box using the following equation:

$$N_{x,y,z,t+1} = N_{x,y,z,t} + \frac{dN}{dt} \times \frac{\Delta t}{V} \tag{1}$$

where $\frac{dN}{dt}$ is the rate of production number (the rate of ship aerosol emissions) in s$^{-1}$, $\Delta t$ is the model time step (75s), and $V$ is the grid box volume. This describes the aerosol concentration added to the entire grid box per time step. At this model resolution, ship tracks form at a minimum width of 1.5km. Typical ship tracks have a width of on the order of 10km (Durkee et al., 2000), therefore this model resolution should be suitable to capture the horizontal extent.

Realistic emissions of condensation nuclei (CN) from ships fall in the range $10^{16}$ to $10^{18}$ kg/s (Taylor and Ackerman, 1999;

Hobbs et al., 2000; Berner et al., 2015). In this work we use $5 \times 10^{14}$ due to issues with activation schemes at very high aerosol concentrations. This approximately matches the $N_d$ perturbation seen in the ship tracks (see Section 3.1). We add the aerosol into the accumulation soluble mode with a mass $3 \times 10^{-18}$ kg, the same as background aerosol, so that all aerosols within the simulation are consistent. This is done to simplify simulations and isolate causal aerosol effects in our analysis.





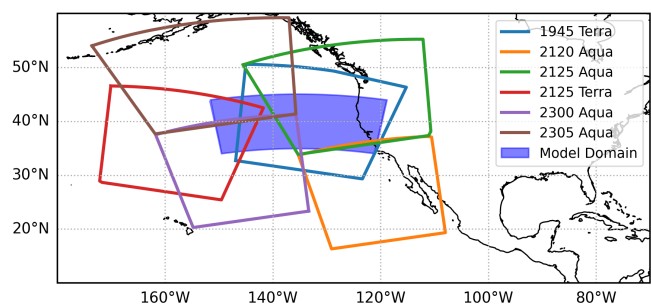

**Figure 2.** Granules from Aqua and Terra overpasses on 12th July 2018 that intersect with the model domain of this study.

## 2.5 Satellite data

To constrain our model simulations, we use observations from NASA's Aqua and Terra satellites to compare to our model output. We use Level-2 Collection 6.1 data from the Moderate Resolution Imaging Spectroradiometer (MODIS; Platnick et al., 2017) to obtain LWP and derive $N_d$ following Quaas et al. (2006).

We collocate Aqua and Terra overpasses the occur during our simulation with our model domain, and only consider images the contain any part of a ship track, obtaining 6 snapshots of our domain, as shown in Fig. 2. Model data is output hourly, therefore we compare satellite data to model output from the nearest time step. When model data is compared to satellite data, the model output is masked only to consider the region contained within the associated satellite snapshot.

Observations of precipitation in the domain are confirmed with overpassed from the Cloud Profiling Radar (CPR) onboard CloudSat (Stephens et al., 2008)) during the simulation run time. We investigate the probability of precipitation across $N_d$-LWP space using observations from the CCCM (CERES–CloudSat–CALIPSO–MODIS) combined product (Kato et al., 2010). We use the CloudSat precipitation flag to define a probability of precipitation as counts of liquid precipitation or drizzle divided by all counts, per $N_d$-LWP bin. Since in 2018 CloudSat was not part of the A-train (and therefore not collocated with MODIS), we use observations from 2007-2011, as in Gryspeerdt et al. (2022).

## 2.6 Ship track locations

We follow a methodology similar to that of Tippett et al. (2024); Gryspeerdt et al. (2021); Manshausen et al. (2022) to obtain our ship track locations. Ship positions (from AIS data) are advected in wind fields over time to obtain a ship track which consists of not only location, but the associated time since that location experienced the aerosol perturbation. This allows the length of a ship track to be treated as a time axis, and discern time evolution after an aerosol perturbation even in snapshot imagery (Kabatas et al., 2013).

In our observations of ship tracks in this case study, the hourly ship locations are advected in ERA5 reanalysis winds at 0.25° resolution and 3 hour intervals (Hersbach et al., 2020) between the surface and the boundary layer top, with vertical motion





within the ship plume following Briggs (1965) (see Tippett et al., 2024 for further details). In model output, the same ship locations are advected instead in the model winds following the same methodology.

Since this study is comparing only 5 ship tracks between model output and satellite data, it is essential that the exact ship track location is logged as to properly quantify the aerosol effects. We apply small corrections to our model and satellite output track locations from the above methodology to ensure that our track location falls exactly on each visible ship track, for both model output and observations. This is done by hand logging the nearest point of the visible track, perpendicular to the predicted track location. This is not possible in large composite studies which contain thousands of ship tracks, however due to the small number of ships considered in this study it is feasible, and ensures the most accurate measurement of the cloud responses.

Following Tippett et al. (2024), we regrid our datasets to 2D space for each ship track, at each model time step / MODIS snapshot. This 2D space consists of the "time along track" (binned to 1 hourly windows), and the perpendicular "distance away from track" (binned to 1 km). This allows us to investigate exactly how the cloud properties vary with time since aerosol perturbation and distance from the centre of the perturbation.

## 2.7 Quantifying enhancements

A key challenge in this study is determining the most appropriate method for quantifying the impact of ship emissions on clouds across our two datasets: model simulations and satellite observations.

For the UM-CASIM simulations, the impact of ship-emitted aerosols can be directly quantified by comparing the enhancement in cloud properties between the *ships* run and the *control* run. To mitigate stochastic noise between model runs, we apply Gaussian smoothing to the model fields (smoothing with a Gaussian kernel with standard deviation of 0.75 km) before calculating the percentage difference. This approach allows for a precise estimation of aerosol-induced changes while minimising the influence of small-scale variability in between model runs.

In contrast, defining a control region in observational data is less straightforward (Christensen et al., 2022). Traditional observational studies of ship tracks typically identify an "unpolluted" reference region located approximately 30 km perpendicular to the ship track (Manshausen et al., 2022). This region serves as a control from which the percentage enhancement in cloud properties is calculated. However, this methodology presents potential biases. Specifically, non-linear background gradients can introduce systematic errors in the estimated aerosol effect (Tippett et al., 2024). Additionally, in scenes containing multiple ship tracks, this approach becomes increasingly uncertain. For instance, in our case study, many "outside track" regions that are used as controls inadvertently overlap with other ship tracks. Consequently, the percentage enhancement derived from such comparisons likely underestimates the true aerosol impact (Yuan et al., 2025).

Despite these limitations, this method remains the most viable approach given the constraints of observational data. In Fig. S3 we demonstrate that both methods for calculating model enhancements produce the same results (to within -15% and -10% for $N_d$ and LWP enhancements in the first 15 hours of ship tracks, respectively). This is due to the model background being relatively clean, and therefore is a good representation of the control cloud since there is not much background variability. In




this study, we choose to employ different methodologies for the model and observational datasets when calculating percentage

enhancements, since in this clean case it does not impact the results.

## 3  Results

### 3.1  Activation of aerosol

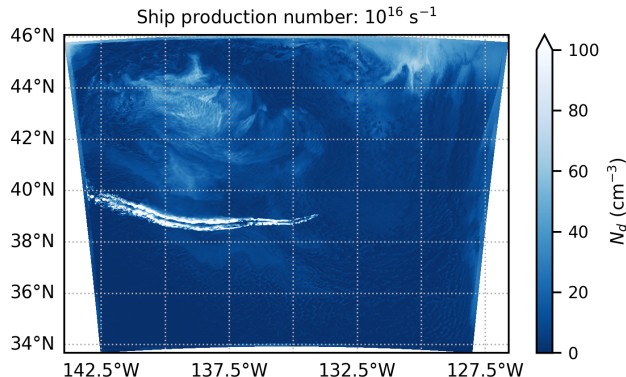

**Figure 3.** Example of how realistic ship emissions ($10^{16}$) produce a 'split' ship track due to non-monotonic activation parameterisation from Abdul-Razzak et al. (1998).

Using Equation (1), and a realistic ship production number of $10^{16}$ s$^{-1}$, we obtain an aerosol perturbation of roughly 20,000 cm$^{-3}$. This is realistic given in-situ observations of ship tracks (Hobbs et al., 2000; Noone et al., 2000). However, with these

values, our simulated ship tracks exhibit bizarre 'split' behaviour, with no cloud droplets activated inside the centre of the track where the aerosol concentrations are highest (see Fig. 3).

This unphysical split-shiptrack effect is due to non-monotonic behaviour of the Abdul-Razzak and Ghan (ARG) activation parameterisation scheme with increasing aerosol concentrations, beyond some critical aerosol concentration (dependent on temperature, pressure and updraft velocity) (Abdul-Razzak and Ghan, 2000). To demonstrate this, we utilise *pyrcel* (Rothenberg

and Wang, 2016), a python package for implementing simple adiabatic cloud parcel model, to isolate solely the impact of the activation parameterisation.

We plot both the ARG and NS (Nenes and Seinfeld, 2003) parameterisations as a function of aerosol concentration in Fig. 4a, using values of temperature, pressure and updraft velocity that are representative of our domain. We find that with using our realistic ship production numbers ($10^{16}$ s$^{-1}$) the aerosol concentrations (20,000 cm$^{-3}$) are beyond the critical concentration of

ARG, producing the unrealistic split ship tracks. NS, whilst not containing this non-monotonic behaviour, would contain a large over estimation of cloud droplets at the realistic ship production number. This detail of the ARG activation parameterisation has be documented in Connolly et al. (2014), and emphasises that commonly used aerosol activation parameterisations are not suitable for MCB applications.





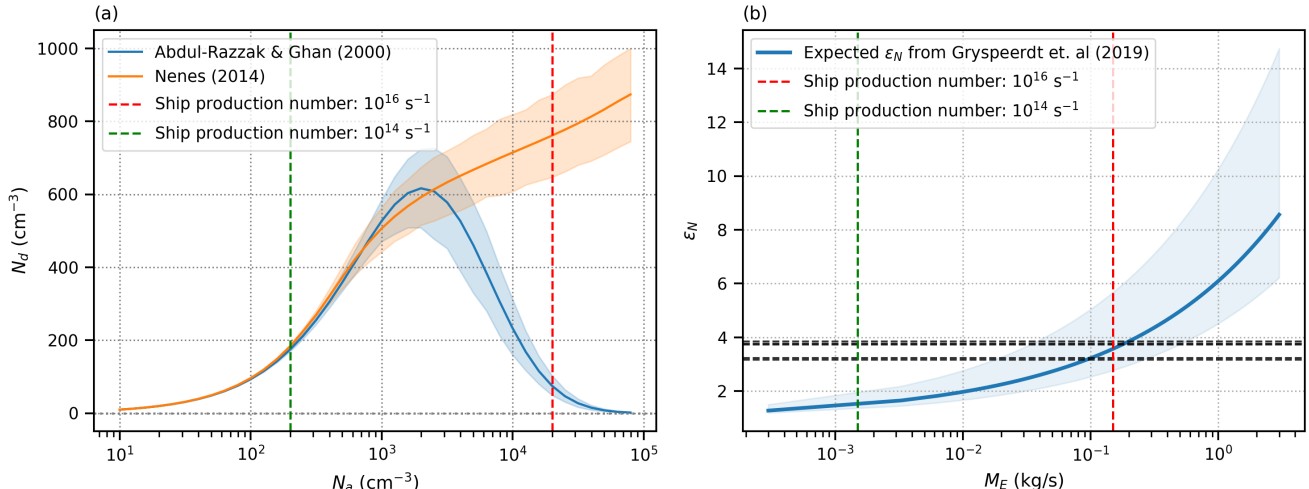

**Figure 4. (a)** Number of aerosol particles activated to cloud droplets in different activation parameterisations calculated in a simple adiabatic cloud parcel model. Confidence intervals given by the range of temperature, pressure and updrafts simulated in our case study. **(b)** The expected enhancement as a function of emissions, based on equation (1) from Gryspeerdt et al. (2019b). Vertical lines are the emissions from different ship production numbers. Horizontal lines are the enhancements produced from our simulations for each individual ship. Confidence interval is given by the range of background droplet number concentrations in our simulation.

Reduction of our ship production number to $5 \times 10^{14}$ s$^{-1}$, produces only a 200 cm$^{-3}$ aerosol perturbation, but recovers $N_d$ that match much closer to observations. This artificial reduction in ship production number is necessary to obtain more realistic ship tracks (without the splitting effect), but we must verify whether or not the resultant enhancements in these ship tracks is as expected based on observations. Using equation (1) from Gryspeerdt et al. (2019b), we calculate expected enhancements ($\epsilon_N$) based on the background $N_d$ as a function of aerosol mass emission rates

$$\epsilon_N = \frac{M_E^\gamma}{\alpha + \beta N_{cln}} + 1 \qquad (2)$$

where $M_E$ is the mass emission rate (in kg s$^{-1}$), $N_{cln}$ is the unperturbed background $N_d$, and the constants $\alpha$, $\beta$, and $\gamma$ are 0.1041, 0.0038 and 0.36 as obtained from the fit in Gryspeerdt et al. (2019b). Using our ship production of numbers of $10^{16}$ s$^{-1}$ and $5 \times 10^{14}$ s$^{-1}$, and a mean aerosol mass of $3 \times 10^{-18}$ kg, we obtain mass emissions rates of $1.5 \times 10^{-1}$ kg s$^{-1}$ and $1.5 \times 10^{-3}$ kg s$^{-1}$, respectively.

We find that the model enhancement obtained from our reduced emissions are consistent with what we would expect from the real emissions (Fig. 4b), despite the emissions being unrealistically low. Since the enhancements obtained are as expected based on observations, we assume that this reduced ship production number is sufficient to drive this realistic test case, and suitable for evaluation of the model's process representation.





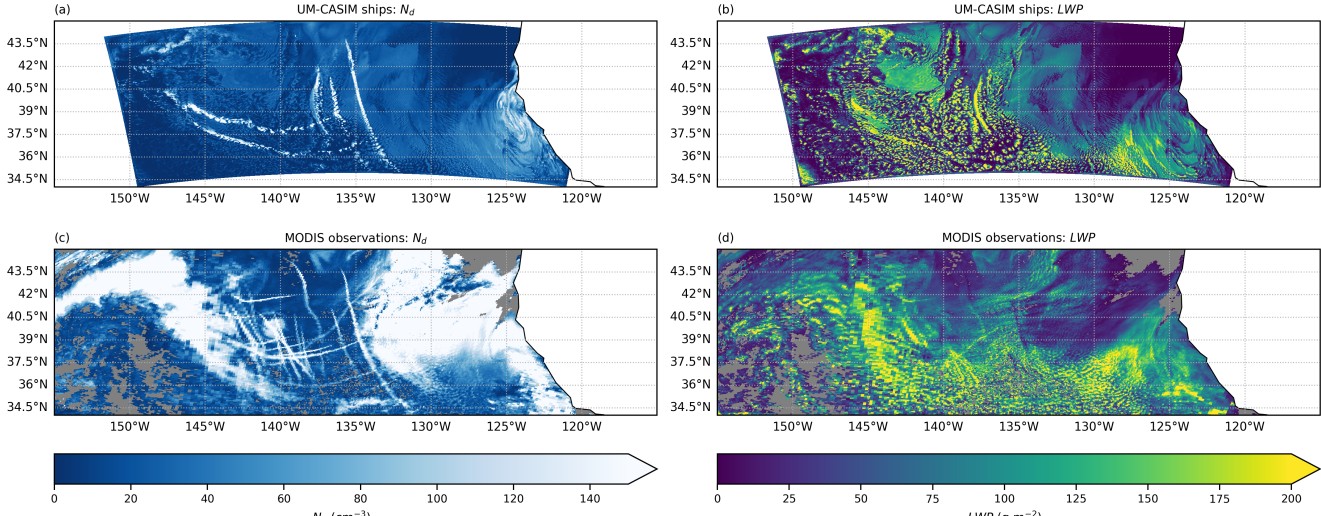

**Figure 5.** Modelled and Observed $N_d$ and LWP for 12th July 2018 ship tracks. Plotted is a composite of the Aqua and Terra overpasses from Fig. 2, co-located with model nearest output time steps.

## 3.2 Comparison to observations

As outlined in Section 2.1, we evaluate our simulations against three criteria in order to determine the realism of our simulations, and the ability of our model to accurately simulate ship tracks / MCB. In the following sections, we present our findings with respect to each of these criteria.

### 3.2.1 In-track values

Fig. 6 provides a comparison between the mean of all MODIS observations of our ship tracks, and the associated modelled ship tracks (at the nearest time step). The inside track values of both $N_d$ and LWP are a close match between model and observations, with a visible peak in the $N_d$ at early times along track, and a slow increase in LWP with time along track which is consistent with the majority of the tracks moving down a LWP gradient.

The mean percentage errors in the inside track values of $N_d$ and LWP, up to 15 hours along the track, are found to be -3.5% and +3.1%, respectively. Therefore, for both $N_d$ and LWP the model reproduces observed values to within +/- 4%, which is well within our uncertainty from the methods (Fig. S3). With respect to our first evaluation criteria, this means that with the artificial reduction in aerosol concentrations necessary for the activation parameterisation to produce realistic shiptracks (see Section 3.1), we are able to successfully reproduce the absolute $N_d$ and LWP values at the location where the aerosol is injected.





**Figure 6. (a)-(d)** Observed $N_d$ and LWP across all observations of our 5 ship tracks, as well as the percentage enhancement of the "inside track" region (defined by <15 km from the centre). Percentage enhancement is calculated from the mean of the region between 30-45 km away from the centre of the track. **(e)-(h)** The same as above, but for modelled ship tracks. The percentage enhancement is instead calculated as the percentage change from the *control* model run. **(i)-(j)** Mean of the observed and modelled $N_d$ in-track values, and enhancements, in the "inside track" region. **(k)-(l)** The same, but for LWP. Whilst inside-track values for both $N_d$ and LWP are similar between observations and model output, enhancements vary significantly due to the definition of "unpolluted" background.



### 3.2.2 Timescales of response

If instead of considering the raw in-track $N_d$ and LWP values, which do not clearly demonstrate the time evolution of the
aerosol effect (since we are not considering a change from an unperturbed state), we calculate the percentage enhancement of
the "inside track" region, compared to the "unpolluted" control cloud.

We find that the initial $N_d$ increase and peak of roughly 300% at 3-4 hours is the same in both model and observations. However, despite the in-track values matching very closely between model and observations, the enhancements vary dramatically
at longer time scales. In Fig. 6j, the enhancement in $N_d$ is much longer lived than that seen in observations, which instead has
returned to the background beyond roughly 15 hours. Since tracks are corrected by hand, this observed lifetime is not a result of
inaccuracies in ship track location prediction. This highlights the importance of considering the change from the unperturbed
state.

Similarly, for the LWP enhancements we see distinctly different responses. In the observations we see the response typically
associated with ship tracks in non-precipitating environments, with a decrease in LWP in the first 5 hours, which then returns
to zero. Conversely, in the modelled ship track we see only a blanket increase in LWP which increases over time. Fig. 6k
demonstrates that the absolute inside track LWP values are similar, therefore this difference in enhancement must stem from
the model representation of the control / background state, and the sensitivity of the model to aerosol.

Overall, we find that evaluating our model against this criteria in this way reveals issues with the lifetime of the aerosol
response, and that current set up and formulation of these model parameterisations is unable to capture the observed lifetime
of the $N_d$ and LWP changes inside these ship tracks. This will have important implications for the radiative forcing from these
ship tracks, since a longer lived aerosol response would prolong the brightening inside the ship track, and therefore would
cause an overestimated cooling effect.

### 3.2.3 Sensitivity to initial condition

In Fig. 7, we split our ship tracks into those that occur in different background conditions, as a means to investigate whether
our model is able to capture varying responses, and to assess the impact on model performance. Since there is precipitation
within the scene, we divide our tracks into precipitating / non-precipitating cases. We use direction of travel as a simple
proxy for "initial condition", since each grouping of ships is always within approximately 100 km of each other, and therefore
experiences largely similar meteorological conditions. Ships A-C follow routes from the coast of California, westwards into
the cleaner ocean (see Fig. 1b). Ships D and E travel towards the coast and are more southerly than ships A-C. Ships A-C travel
through primarily non-precipitating clouds, whereas ships D and E travel through drizzling clouds (at the surface; see Fig. S4).
Dividing our enhancements into these two groupings, we see that the model is sensitive to the initial condition of the cloud
before the ship passes through (Fig. 7).

In non-precipitating conditions (Fig. 7, 1st and 3rd column for $N_d$ and LWP enhancements, respectively) we find similar
enhancements in the model and observations. On short timescales there is better agreement, with observations and model simulations matching within the uncertainty in the first five hours. Following this, we see divergence between the two. This could



**Figure 7.** Percentage enhancements from the "unpolluted" references background state. **(a)-(d)** show observations of ship tracks in non-precipitating (ships A, B and C) and precipitating (ships D and E) conditions for $N_d$ (left) and LWP (right). **(e)-(h)** shows the same but for modelled ship tracks. **(i)-(l)** show the average of the "inside track" region, as defined by the red dashed lines in **(a)-(h)**.





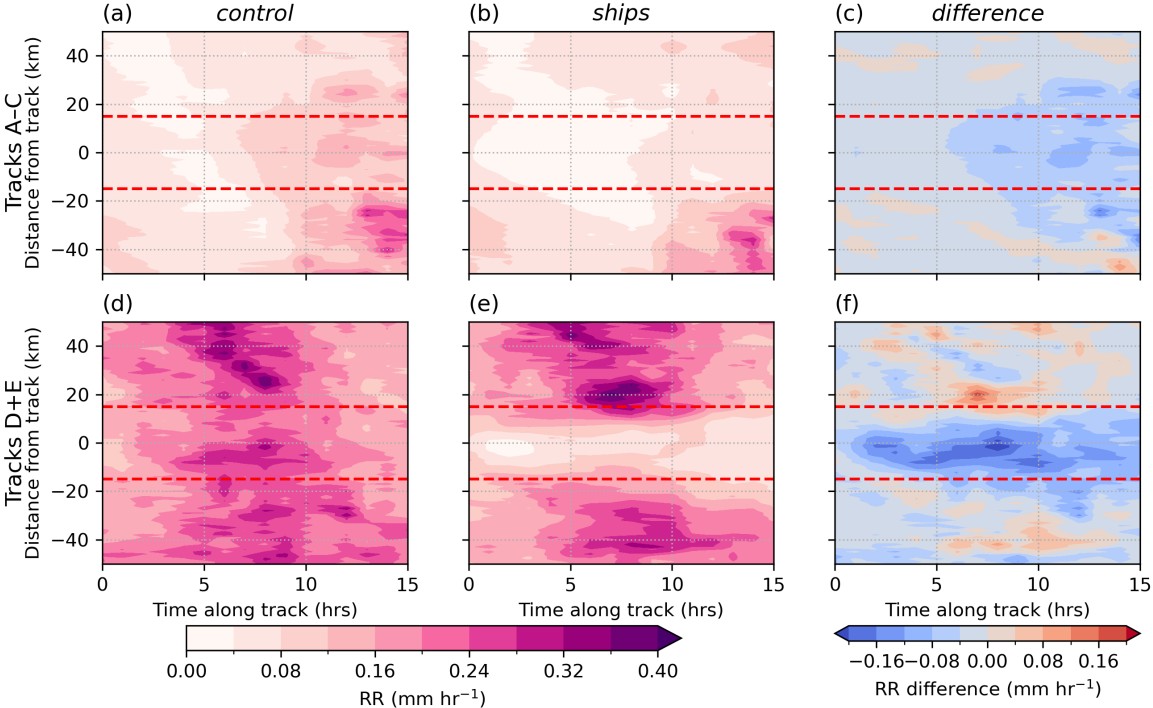

**Figure 8.** Surface rain rate, averaged across two groupings of tracks (ships A-C and D+E), across 24 hours on the 12th July 2018, for the *ships* and *control* runs, as well as the difference between the two. Clear precipitation suppression in the "precipitating" case, with rain rates reduced to almost zero. Some precipitation suppression in the "non-precipitating" case (ships A-C) since there is actually some precipitation towards the end of the model run (see Fig. S4).

be in part due to our tuning of the aerosol emissions to satisfy the non-monotonic behaviour of the activation parameterisation and reproduce correct $N_d$ enhancements on short time scales. We would expect the response from weaker aerosol perturbations to reduce over time, and therefore there must be some process within the model allowing for this increased $N_d$ perturbation lifetime.

Considering the ships that travel through precipitating clouds (Fig. 7 2nd and 4th column for $N_d$ and LWP enhancements respectively), we see significant disagreement between the observations and the model. This suggests that the large disagreement in Fig. 6 is largely due to these ships in precipitating conditions. The $N_d$ enhancement is too large, and shows no sign of diminishing even after 15 hours after the aerosol perturbation. This suggests that the model is not only overly sensitive to aerosol in precipitating conditions, but is not effective enough at removing aerosol from the cloud either. Additionally, the

LWP response is a different sign between model and observations. This suggests that the model is far too keen to suppress precipitation, leading to large increases in LWP that are not observed in the satellite imagery.



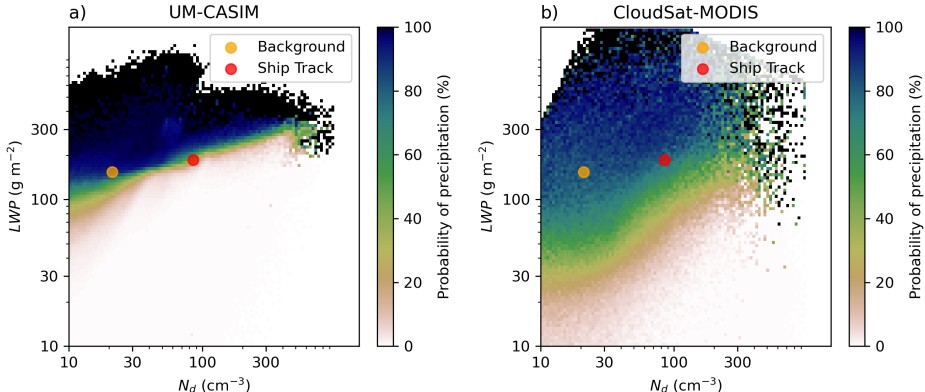

**Figure 9.** The probability of precipitation (PoP) in $N_d$-LWP space, in UM-CASIM simulations compared to CloudSat-MODIS collocated observations for the same domain. UM-CASIM has much faster transition from precipitating to non-precipitating with changes in $N_d$ than observed. The orange point is the background (*control*) $N_d$ and LWP for our precipitating ship (D-E), and the red point is the $N_d$ and LWP inside the ship tracks, averaged over the simulation run time.

In Fig. 8, we show the surface rain rates inside our ship tracks, compared to the control run. Here, we see that for tracks D and E (which occur in more precipitating environments), the precipitation is essentially completely shut off which is likely a far too strong response, since precipitation in high $N_d$ conditions does occur (Gryspeerdt et al., 2022).

We demonstrate this in Fig. 9, where the probability of precipitation (PoP) is plotted in $N_d$-LWP phase space for our model simulation and colloacted CloudSat-MODIS observations (following a similar methodology to Gryspeerdt et al., 2022). The very sharp gradient in PoP with increases in $N_d$ demonstrates the high-sensitivity of the autoconversion scheme to aerosol. In our precipitating case, the addition of ship aerosol moves our "background" point in $N_d$-LWP phase space just over the sharp boundary in PoP to our "ship track" point. This explains the complete precipitation suppression in these cases, since this

gradient is so sharp. In observations, there would perhaps be a small decrease in PoP, but not to the same extent.

In our simulations, we do see that there is some sensitivity to the initial conditions since we obtain different responses in precipitating / non-precipitating cases. We can conclude that the model is more suitable for simulations of high-concentration aerosol perturbations in non-precipitating conditions, at least for up to 5 hours. However in the case of precipitating clouds, current model parameterisation formulations are not suitable to accurately simulate these scenarios, and the lifetime of the

response is far too long lived.

## 4 Discussion

In this study, we produce simulations of ship tracks using a regional model and double moment microphysics scheme, exhibiting its capabilities and limitations in simulating high-concentration aerosol interactions with clouds. We evaluate our case





study against three criteria, which are designed to ensure ship track simulations contain an accurate representation of cloud
processes, and would be a useful tool for MCB research.

With some tuning of parameterisations, we reproduce realistic absolute values of $N_d$ and LWP inside simulated ship tracks,
and find that in non-precipitating conditions the enhancement in $N_d$ and LWP from the control "unpolluted" cloud is well
modelled for the first 5 hours. However, beyond this, the lifetime of the response is too long lived. This is driven by the
representation of parameterisations at very high aerosol concentrations.

### 4.1    Definition of the "control" region.

In order to quantify the enhancement of cloud properties inside ship tracks from their background state, we must first define
this unperturbed "control" cloud.

In model simulations, we are able to define this region from our *control* simulation, where there are no ship emissions and
we can directly measure the unperturbed cloud. In observations, however, we do not have a control run, and therefore region
is more complex to define. Typically, we consider the region neighbouring the ship track as being representative of what the
cloud would have looked like if there was no ship present, and use this to infer the aerosol effect on the cloud.

Whilst this methodology is useful to predict the control cloud, it is not perfect. Ship tracks often occur in large groupings
and have considerable overlap, which may distort the ability to define the "clean" background and make it difficult to observe
aerosol effects at longer lifetimes. This introduces a source of uncertainty when comparing enhancements in Fig. 6 and Fig. 7.
We compare these two different methodologies for defining the control in our model simulation in Fig. S3. The first method is
by using the model control run, and the second is by using the "outside" ship track region (as is typically done for observations).
In our model simulation, we find that this introduces 10% and 15% underestimation in the $N_d$ and LWP enhancements,
respectively. However, since our model simulation is very clean, and does not contain the many other ship tracks seen in
observations, this gives a lower bound on the expected underestimation from defining the control region using the neighbouring
clouds.

Essentially, with fewer background sources of aerosols, it is easier to define the "unperturbed" cloud using the surrounding
region. When there are more ships, or other sources of aerosols, this definition becomes more difficult and becomes a greater
source of uncertainty. Other studies have explored alternative methodologies of gaining insight into the unperturbed cloud in
observations (such as through the use of ML algorithms; Diamond et al., 2020; Chen et al., 2022; Diamond, 2023), however
will still struggle with increasingly complex and polluted backgrounds.

This has important implications for the design of field experiments of MCB. In order to isolate the aerosol effect with
greatest certainty, a relatively simple background is necessary. If there are many overlapping ship tracks / interacting sources,
this definition of the control becomes more nuanced, and additional uncertainty is introduced. If field trials were to occur for
MCB, the easiest, and best attribution of the aerosol effect could be made if ships tracks were not allowed to interact, and the
background was not too heavily polluted with sources that were difficult to define.





## 4.2 Suitability of parameterisations for high-concentration aerosol perturbations

We find that the commonly used activation parameterisation by Abdul-Razzak and Ghan (2000) is inadequate to simulate aerosol-cloud interactions at very high aerosol concentrations. In practical terms, this necessitates an artificial reduction in aerosol input to obtain ship track structures that resemble those seen in satellite imagery. While this workaround may yield visually plausible results in non-precipitating conditions, it undermines the physical realism of the simulation at long timescales.

Model-observation discrepancies become more pronounced in precipitating environments. Under such conditions, it is possible that the autoconversion parameterisation of Khairoutdinov and Kogan (2000, KK00), which governs the conversion of cloud droplets to rain, is too sensitive to increases in $N_d$. This scheme tends to suppress production of rain droplets too readily when the cloud droplet number concentration ($N_d$) is high - as demonstrated by the complete shut off of precipitation inside these ship tracks that form in precipitating scenes (see Fig. 8).

In Fig. 9, the PoP is almost binary in our simulation, with either 100% chance of precipitation above some threshold $N_d$-LWP line, and almost 0% chance below it. In observations, there exists a much wider space in which precipitation can occur. This means that for any conditions close to this threshold line in $N_d$-LWP in our simulation, an increases of $N_d$ (such as through the injection of ship emissions), will move us beyond this threshold and decreases the PoP too extremely.

Consequently, the model predicts unrealistic suppression of precipitation, resulting in overestimation of the ship track lifetime and, by extension, the radiative cooling associated with the perturbation. This is not surprising, since the parameterisation from KK00 was only designed and tested on mid-latitude and extratropical stratocumulus layers under typical meteorological conditions, with their "polluted" case only having an $N_d$ of 175 cm$^{-3}$. This is aligned with the findings of Jing and Suzuki (2018) with respect to GCM parameterisations, where precipitation inhibition can lead to a wet scavenging feedback which can increase aerosol loading somewhat non-physically. In Gryspeerdt et al. (2022), it is found that KK00 completely suppresses precipitation at high $N_d$, despite there still being some probability of precipitation if LWP is high. KK00 is likely not well-suited to simulate very high aerosol concentration perturbations, as it tends to easily suppress precipitation - even in conditions where precipitation could still occur.

In their designed configurations, these parameterisations limit the ability of models to provide meaningful insight into high concentration aerosol-cloud interactions over longer timescales or under precipitating conditions. This has important implications with respect to climate modelling, where high-resolution global cloud resolving models are being employed. Additionally, for simulations aimed at informing MCB strategies, it is particularly critical that parameterisations are suitable for high-aerosol scenarios. Without modifications, models will likely overstate the cooling impacts, leading to inaccurate assessments of MCB efficacy. Improved autoconversion schemes that remain valid at high $N_d$, along with more robust activation parameterisations that avoid the need for artificial aerosol reductions, are essential for advancing the realism of such simulations.

## 5 Conclusions

Accurate model representation of aerosol-cloud processes is essential for reducing uncertainty in climate projections and simulating the impacts of marine cloud brightening (MCB). Ship tracks provide a "natural experiment" for isolating the aerosol





effects on clouds, and gaining information about the time evolution of the cloud response. In order to evaluate the model
representation of these processes, we simulate a real case of ship tracks observed off the coast of California on 12th July 2018,
using an experimental configuration of the Met Office's UK weather forecasting model.

We evaluate the performance of our model against three criteria, aiming to produce simulations of ship tracks which are a
useful tool for investigating the impacts of MCB. We assess the model's ability to reproduce observed relative and absolute
changes in cloud droplet number concentration ($N_d$) and liquid water path (LWP), using MODIS satellite retrievals for valida-
tion. The ship tracks occur within a marine stratocumulus deck under very clean background conditions, including pockets of
drizzling cloud.

We find that the activation parameterisation of Abdul-Razzak and Ghan (2000, ARG) is not well-suited for simulating high
aerosol concentrations typical of ship plumes. In particular, ARG exhibits non-monotonic behaviour at elevated concentrations,
leading to non-physical results unless aerosol input is artificially reduced by a factor of 20. This adjustment yields raw $N_d$ and
LWP values within ship tracks that closely match observations (Fig. 6).

However, analysis of raw values inside ship tracks does not tell us about the time evolution of the aerosol effect, or how long
it takes the clouds to return to the unperturbed state. In the model, this is easy to define through the use of a *control* simulation.
In the observations, we consider the region just outside of the ship track as a representative control region. This highlights the
usefulness of ship tracks in observing aerosol-cloud interactions, as it provides us with this control region that is otherwise so
hard to define. When considering the enhancements from the control state, we find that the model fails to accurately simulate
the time evolution of the response that is seen in observations. Specifically, model enhancements in $N_d$ persist too long, and the
LWP response diverges in sign from observations. On average, across five simulated ship tracks, the model does not capture
the correct time evolution of the aerosol effect. There is some uncertainty (roughly 10%) introduced by the definition of the
control state in the observations, since it is difficult to define the unperturbed cloud when there are many overlapping ship
tracks. Additionally, some of this discrepancy could be attributed to a distortion in the time evolution due to the tuning of the
activation parameterisation (such that physical ship tracks were produced).

We further classify ship tracks by background precipitation conditions to better understand this discrepancy. In non-precipitating
environments, the model captures the initial enhancement in cloud properties but still exhibits a response that is too long lived
compared to observations (Fig. 7i,k). This suggests that while the initial tuning of the activation parameterisation produces
realistic tracks, it may distort the subsequent cloud evolution, and points to the broader need for improved activation schemes
capable of handling extreme aerosol conditions, especially in contexts such as MCB (Connolly et al., 2014).

In precipitating environments, model-observation disagreement is more pronounced, which is critical for studies of cloud
adjustments to aerosols. Model $N_d$ enhancements are significantly overestimated and show no decay over time, whereas ob-
servations indicate a return to background values within 15 hours (Fig. 7j). Moreover, the model shows a strong positive LWP
response, while observations show a weakly negative one. This suggests that the model either (a) overestimates the sensitivity
of precipitation suppression to aerosol, (b) misrepresents background precipitating conditions, or (c) cannot be compared to
potentially unrepresentative observations.

This has particular relevance for solar radiation management proposals such as MCB, and aerosol forcing estimates from high resolution models. Current model parameterisations may cause an overestimation in the cloud responses to aerosol and predict an unrealistically strong cooling effect, especially under precipitating conditions. Any application of these models to unobserved phenomena like MCB must be grounded in rigorous comparison with real-world observations, and further work is necessary in modifying parameterisations such that they are more suitable for high-concentration aerosol perturbations.

*Code and data availability.* MODIS data used in this work were acquired from the Level-1 and Atmosphere Archive and Distribution System (LAADS) Distributed Active Archive Center (DAAC) (https://ladsweb.modaps.eosdis.nasa.gov, LAADS DAAC, 2021; Platnick et al. 2017). The ERA5 data are from the Copernicus Climate Change Service (C3S) Climate Data Store (CDS) (https://doi.org/10.24381/cds.adbb2d47, Copernicus Climate Change Service, 2023; Hersbach et al. 2020). Ship AIS data were obtained from exactEarth. exactEarth data are not publicly available; however, they can be accessed via a paid subscription: https://www.marinetraffic.org/exactearth (last access: 29 November 2024). Code and data from this analysis will be made available upon publication, via Zenodo.

*Author contributions.* All authors contributed to the design of the study. PRF provided access to the Met Office model and Monsoon, and supported AT in the set up and running the simulations. AT performed the model output and observational analysis, with EG providing observations data for Fig. 9. PRF and EG assisted with the interpretation of the results. AT drafted the manuscript, and PRF and EG provided comments and suggestions.

*Competing interests.* No competing interests are present

*Acknowledgements.* We acknowledge use of the Monsoon2 system, a collaborative facility supplied under the Joint Weather and Climate Research Programme, a strategic partnership between the Met Office and the Natural Environment Research Council. Anna Tippett and Edward Gryspeerdt acknowledge funding from the Horizon Europe programme (project CERTAINTY – Cloud–aERosol inTeractions & their impActs IN The earth sYstem; grant agreement no. 101137680) and a Royal Society University Research Fellowship (grant no. URF/R1/191602). Paul R. Field acknowledges funding from ...



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
