# Peer review of "Evaluating simulations of ship tracks in a high-resolution model"

_EGUsphere, 2025_

## Referee Comment (RC1)

Anna Tippett1, Paul R. Field2,3, and Edward Gryspeerdt1 Evaluating simulations of ship tracks in a high-resolution model https://doi.org/10.5194/egusphere-2025-3877

**Summary**

This research focusses on the evolution of simulated ship tracks, examining the sensitivity of the duration of the ship track to precipitation intensity within the marine atmospheric boundary layer clouds. The simulated ship tracks located within regions of classic stratocumulus cloud decks (closed mesoscale cellular convection) are found to persist for hours to days, in good agreement with observations. The simulation of ship tracks over the central Pacific, within more broken MABL clouds, potentially open MCC, similarly persist in time, which is not in good agreement with observations. Here the ship tracks are observed to be relatively short-lived. The hypothesis is that simulations are overly sensitive to the aerosol loading of the ship tracks, shutting down precipitation and allowing the ship tracks to persist for too long. Effectively the simulations are too sensitive to the 2nd aerosol indirect effect (i.e., the Albrecht effect.).

The paper is well-motivated and timely, building on the increasing interest in marine cloud brightening. It falls well within the scope of ACP and will be of interest to the broader community.

**Recommendation**

This work needs major revisions, as identified below. While I generally accept their hypothesis, I find their premise to be overly simplistic and their analysis needs to be more rigorous.

**Major Comment 1:**

The basic premise is that the only difference between the Eastern Pacific ship tracks (A-C) and the Central Pacific ship tracks (D & E) is the intensity of the precipitation, hence the difference in the longevity of the ship tracks is strictly a function of the precipitation. Both the satellite observations and simulations suggest that these two regions have vastly different types of MABL clouds with classic closed MCC in the Eastern Pacific and broken cumulus in the Central Pacific, potentially transitioning to trade cumulus or potentially open MCC. In terms of MABL clouds, you are comparing apples to oranges. There are differences in cloud fraction, MBL/cloud top height, entrainment from the free troposphere, large scale subsidence, wind speed, estimated inversion strength, and sensible and latent heat fluxes off the ocean. Do we know if the Central Pacific MABL is decoupled or not? How do we know that the rate of collision & coalescence removing aerosols within CASIM isn't sensitive enough to turbulence in the MABL?

To simply attribute all the differences in the persistence of the ship track to precipitation alone, is overly simplistic.

While I find this a major concern, it can readily be addressed with a more comprehensive, robust discussion. Be upfront with the limitations of the analysis.

**Major Comment 2:**

"Observations of precipitation in the domain are confirmed with overpassed (sic) from the Cloud Profiling Radar (CPR) onboard CloudSat (Stephens et al., 2008)) during the simulation."

Are you really trying to say that since CloudSat (which CloudSat product, the 2C-column-precip or the 2C-rain-profile?) recorded precipitation somewhere along it's overpass, that the simulated precipitation is reasonable?

In all sincerity, I was inclined to recommend rejecting the manuscript on that single sentence alone.

This is a major weakness in research, the simulated precipitation has not been evaluated in any way. Given all the challenges we have in estimating precipitation in shallow convection over the remote ocean and the significant differences commonly found between various precipitation products, both from reanalyses and satellite-based products, it needs to be shown that simulated precipitation has some measure of skill.

I strongly recommend that a section on the evaluation of the simulated precipitation be added to this manuscript. Ideally, the evaluation would be made against both ERA5 and GPM-IMERG for the full domain, commenting on the level of skill for both the classic stratocumulus over the Eastern Pacific and the broken cumulus of the Central Pacific. In addition, an evaluation should be made along the CloudSat overpass, against both the 2C-RP and 2C-CP products.

**Major Comment 3:**

There appears to be a difference between the simulations and the satellite imagery in the cloud structure/morphology in the Central Pacific. This difference is evident in Figure 5. Do ship tracks D & E pass through this cloud field? If so, please comment on how this may affect the analysis.

**Minor comments**

Line 32: "large-scale" is ambiguous to me in this sentence.

- Line 32: The whole sentence confuses me. Why are you isolating large-scale turbulence here? Are you saying the large-scale turbulence within coarse resolution CGMs are used in activation of CCN in boundary layer clouds?
- Line 34: "certainty"? We will never have certainty. But we can have some measure of confidence.
- Line 47: This sentence is out of place to me. The changes to the LWP happen after the suppression of precipitation (Albrecht effect), which is in the next sentence.
- Line 72: "short timescales" is ambiguous
- Line 80: The simulated ship locations must be prescribed whether from an actual ship or not.
- Line 115: It would be nice to add text stating that the domain covers the Eastern and Central Pacific, off the coast of California.
- Line 120: Do you need four references here?
- Line 175: Is 'constrain' the right word here? You aren't assimilating any of the satellite observations, are you?
- Line 182: Which CloudSat product?
- Line 183: I skimmed through Kato et al. (2010) but could find no mention of a precipitation product being produced, nor LWP or Nd. Is this reference correct? If so please add the detail necessary to from the referenced material to the products used in this paper.
- Line 190: Ship plumes are advected in wind fields, not positions.
- Line 254: Should you also define  $e_L$  here?
- Figure 1: This may work better if you present the control (left column) and the difference (right column.
- Figure 9b: Is this for the full four-year time period, as stated in line 186? If so, why is it meaningful to compare a four-year average, that contains winter and summer, against a simulation of a single day during July? Please address this issue.
- Final comment: This research focusses the impact of aerosol loading (the ship track) on the clouds and precipitation. It doesn't explore the processes where clouds and precipitation impact the aerosol loading, i.e., collision-coalescence and wetdeposition. It may strengthen your hypothesis to consider this in the discussion.
  - Kang, L., Marchand, R., Wood, R. & McCoy, I. L. Coalescence scavenging drives droplet number concentration in Southern Ocean low clouds. *Geophys. Res. Lett.* **49**, e2022GL097819 (2022).
  - Terai, C., Bretherton, C., Wood, R. & Painter, G. Aircraft observations of aerosol, cloud, precipitation, and boundary layer properties in pockets of open cells over the southeast Pacific. *Atmos. Chem. Phys.* **14**, 8071–8088 (2014).

Wood, R., Leon, D., Lebsock, M., Snider, J. & Clarke, A. D. Precipitation driving of droplet concentration variability in marine low clouds. *J. Geophys. Res.: Atmos.* **117**, D19210 (2012).

Alinejadtabrizi, T., Lang, F., Huang, Y. *et al.* Wet deposition in shallow convection over the Southern Ocean. *npj Clim Atmos Sci* **7**, 76 (2024). https://doi.org/10.1038/s41612-024-00625-1